# Variations in the Abortive HIV-1 RNA Hairpin Do Not Impede Viral Sensing and Innate Immune Responses

**DOI:** 10.3390/pathogens10070897

**Published:** 2021-07-15

**Authors:** Melissa Stunnenberg, John L. van Hamme, Atze T. Das, Ben Berkhout, Teunis B. H. Geijtenbeek

**Affiliations:** 1Amsterdam Institute for Infection & Immunity, Department of Experimental Immunology, University of Amsterdam, Amsterdam UMC, Meibergdreef 9, 1105 AZ Amsterdam, The Netherlands; m.stunnenberg@amsterdamumc.nl (M.S.); j.l.vanhamme@amsterdamumc.nl (J.L.v.H.); 2Laboratory of Experimental Virology, Amsterdam Institute for Infection & Immunity, Department of Medical Microbiology and Infection Prevention, University of Amsterdam, Amsterdam UMC, Meibergdreef 9, 1105 AZ Amsterdam, The Netherlands; a.t.das@amsterdamumc.nl (A.T.D.); b.berkhout@amsterdamumc.nl (B.B.)

**Keywords:** human immunodeficiency virus 1, abortive HIV-1 RNA, viral sensing, TAR hairpin, sequence variation, thermodynamic stability, TAR RNA structure, pattern recognition receptors, antiviral immunity

## Abstract

The highly conserved trans-acting response element (TAR) present in the RNA genome of human immunodeficiency virus 1 (HIV-1) is a stably folded hairpin structure involved in viral replication. However, TAR is also sensed by viral sensors, leading to antiviral immunity. While high variation in the TAR RNA structure renders the virus replication-incompetent, effects on viral sensing remain unclear. Here, we investigated the role of TAR RNA structure and stability on viral sensing. TAR mutants with deletions in the TAR hairpin that enhanced thermodynamic stability increased antiviral responses. Strikingly, TAR mutants with lower stability due to destabilization of the TAR hairpin also increased antiviral responses without affecting pro-inflammatory responses. Moreover, mutations that affected the TAR RNA sequence also enhanced specific antiviral responses. Our data suggest that mutations in TAR of replication-incompetent viruses can still induce immune responses via viral sensors, hereby underscoring the robustness of HIV-1 RNA sensing mechanisms.

## 1. Introduction

All human and simian immunodeficiency virus (HIV and SIV) transcripts carry a hallmark at their 5′ end: the structured trans-acting (TAR) RNA response element [1]. In HIV type 1 (HIV-1), TAR is stably folded into a 58 nucleotide (nt)-long hairpin structure that is involved in many steps of the viral replication cycle, such as packaging of genomic HIV-1 RNA, reverse transcription initiation and translation [2,3,4]. The best-characterized function of the TAR hairpin is its role in boosting viral transcription via interactions with the transactivator protein Tat [5]. The TAR hairpin presents a highly conserved three-nt-long pyrimidine bulge and a six-nt-long loop to which Tat and cellular kinases bind to initiate Tat-mediated *trans*-activation of the HIV-1 long terminal repeat (LTR) promoter and thus transcription of the provirus [6,7,8,9]. The TAR sequence and structure are highly conserved among different virus isolates, likely reflecting its essential role in viral replication, and mutations in TAR usually render the virus replication-incompetent [3,4,10,11]. In contrast to supporting viral propagation, TAR also triggers antiviral responses [12,13,14,15]. Whether mutations that affect the structured nature of the TAR hairpin affect its ability to induce antiviral immunity remains largely unclear.

Upon transcription initiation, RNA-polymerase-II (RNAP II)-mediated transcription along the HIV-1 proviral DNA genome is halted after 58 nts, hereby generating prematurely aborted HIV-1 RNA transcripts that can adopt the complete TAR hairpin structure (abortive HIV-1 RNA) [2]. These short abortive HIV-1 RNA transcripts contain the 5′cap structure but lack the 3′-poly A tail and are recognized by cytosolic pattern recognition receptors (PRRs) [12,16]. In the cytoplasm, abortive HIV-1 RNA transcripts are sensed by DEAD box RNA helicase 3 (DDX3) that recognizes abortive HIV-1 RNA via the 5′cap in close proximity to the TAR hairpin structure [12,13,17]. In addition, the PRR protein kinase RNA-activated (PKR) recognizes abortive HIV-1 RNA, but this depends on the double-stranded RNA of the TAR hairpin structure rather than the 5′cap close to the complex secondary hairpin [14,18,19,20,21,22].

Sensing of abortive HIV-1 RNA triggers various responses via either RNA sensor DDX3 [12,13] or PKR [23]. We have previously shown that triggering of DDX3 by synthetic abortive HIV-1 RNA results in the induction of potent *IFNB* and interferon-stimulated gene (ISG) transcription, a full antiviral type I interferon (IFN) response capable of limiting HIV-1 replication in vitro [13]. In addition, we have shown that PKR-dependent sensing of abortive HIV-1 RNA induces NLRP3 inflammasome activation for the production of the pro-inflammatory cytokine interleukin 1 beta (IL-1β) [23]. Little is known about the effect of abortive HIV-1 RNA sequence variation on viral sensing and the subsequent antiviral or inflammatory responses.

Here, we introduced mutations within the aborted HIV-1 TAR RNA to study the effect of its thermodynamic stability and RNA structure on viral sensing by PRRs. Notably, we observed that mutations that increased or decreased thermodynamic stability of the TAR hairpin enhanced viral sensing and antiviral immune responses. These data suggest that viral sensors are able to sense a variety of HIV-1 TAR RNA variants. Moreover, changes in the TAR RNA sequence also enhanced viral sensing. Thus, while virus replication is highly dependent on the integrity and stability of HIV-1 TAR, our study suggests that viral sensors are able to sense many different variations in TAR, underscoring the robustness of the sensing mechanisms for HIV-1 TAR RNA.

## 2. Results

### 2.1. Variation in TAR Sequences Affects RNA Structure and Thermodynamic Stability

Here, we examined the effect of variation in the structure and thermodynamic stability of the TAR RNA hairpin on the induction of antiviral immunity. First, the sequence variation in the TAR motif in natural virus isolates of HIV-1 subtype B was examined. The TAR region of >1600 sequences was assessed, and a consensus sequence was derived (Figure 1A). Minimal sequence variation was detected at nt positions 10, 12, 23, 24, 30, 31, 46, 47, 50 and 51 and likely did not interfere with HIV-1 replication (Figure 1A). Several studies have investigated the effect of introducing more profound variation in the TAR hairpin structure on *trans*-activation, and described that this prevents viral replication [3,24,25]. Here, we investigated the effect of variation in the TAR hairpin structure on viral sensing by PRRs.

We developed synthetic abortive HIV-1 RNA based on the consensus sequence [12,13] (TAR wt, Figure 1A) and designed mutant TAR RNAs to determine the effect of changes in the RNA structure on the potency of antiviral responses (Figure 1B,C and Figure 2A,B). The first set of four mutants was designed to further stabilize the TAR hairpin. This was completed either by removal of the destabilizing 3-nucleotide bulge element on the 5′ side (Del-UCU-22) or introduction of the AGA triplet on the 3′ side (Ins-AGA-22) to allow base pairing with the UCU bulge (Figure 1B and Figure 2A). Indeed, the thermodynamic stability is increased by these mutations from delta G −30.4 kcal/mol to −36.2 and −41.8 kcal/mol, respectively. Next, we changed these mutants further by removing destabilizing elements in the bottom half of the TAR hairpin, either by deletion of the destabilizing unpaired nucleotides (C4 and A16 in the context of Del-UCU-22) or insertion of complementary nucleotides on the 3′ side of the hairpin (G4 and U16 in the context of Ins-AGA-22) (Figure 1B and Figure 2A). Both deletion or insertion of base pairs further stabilized the TAR hairpin to delta G −43.8 kcal/mol for Del-C4-A16-UCU-22 and −54.2 kcal/mol for Ins-G4-U16-AGA-22, respectively.

In addition, we designed mutants in which the hairpin was destabilized. We introduced four clustered mutations on the 5′ (TAR mutant Substitution AUCA-9) or 3′ side (TAR mutant Substitution UGGU-47). These mutations prevented base pairing, leading to a profound destabilization of the lower TAR stem, which decreased TAR stability to delta G −19.1 kcal/mol and −20.5 kcal/mol, respectively (Figure 1C and Figure 2B). Next, we combined these 5′ and 3′ mutations, which restored the stability of the TAR hairpin to almost the TAR wt level (double mutant Substitutions 9 + 47, delta-G −30.3 kcal/mol) but with a different sequence at nt positions 9–12 and 47–50 of the TAR hairpin (Figure 1C and Figure 2B). This strategy is suited to probe the importance of sequence versus structural elements in RNA structures. Thus, although TAR RNA sequences are highly conserved, we here introduced sequence variation in the TAR hairpin to study the effect of sequence elements, structure and thermodynamic stability on induction of antiviral responses.

### 2.2. Deletions in the TAR Hairpin That Enhance Its Thermodynamic Stability Increase Antiviral Immunity

To investigate the effect of TAR RNA stabilization on the induction of antiviral responses, human monocyte-derived dendritic cells (DCs) were treated with TAR wt or mutants with higher stability, and *IFNB* transcription was assessed every 2 h (h) after stimulation. Similar to what was previously reported, TAR wt induced strong *IFNB* transcription, which peaked around 10 h after stimulation [13] (Figure 3A). TAR mutant Del-C4-A16-UCU-22 induced *IFNB* transcription at higher levels than observed for TAR wt, starting at 4 h after stimulation and persisting at peak level (Figure 3A). In contrast, *IFNB* transcription induced by TAR mutant Del-UCU-22 was slightly enhanced compared to TAR wt-induced *IFNB* transcription up to 8 h after stimulation, whereas Ins-AGA-22 and Ins-G4-U16-AGA-22 showed a trend towards decreased *IFNB* transcription compared to TAR wt (Figure 3A,B). Next, we assessed the transcription of interferon-stimulated genes (ISGs) and found that the different TAR mutants induced a similar transcription pattern for *IL27A* as compared to *IFNB* transcription (Figure 3C). TAR mutant Del-C4-A16-UCU-22 significantly increased *IL27A* transcription compared to TAR wt and exceeded the response induced by the positive control LPS (Figure 3C). In contrast, mutant Del-UCU-22 induced similar levels, while mutants Ins-AGA-22 and Ins-G4-U16-AGA-22 induced lower levels of *IL27A* transcription compared to the TAR wt (Figure 3C). Furthermore, all TAR mutants induced *APOBEC3G* (*A3G*) and *ISG15* transcription; however, these responses were similar to the TAR wt-induced responses and were not further enhanced (Figure 3C).

Next, we examined the effect of TAR RNA structure stability on the transcription of interferon-regulatory factor 3 (*IRF3*) and *IRF7*. TAR wt induced *IRF3* transcription that peaked at both 4 h and 10 h after stimulation (Figure 3D). Similar to the observations for *IFNB* transcription, the TAR mutant Del-C4-A16-UCU-22 enhanced *IRF3* transcription compared to TAR wt-induced responses at 4 h (Figure 3D,E). In addition, the TAR mutant Ins-AGA-22 showed a trend towards increased *IRF3* transcription, while the Del-UCU-22 and Ins-G4-U16-AGA-22 mutants did not affect *IRF3* transcription compared to TAR wt-induced responses (Figure 3D,E). No effects were seen for *IRF7* transcription upon stimulation with the TAR mutants compared to the TAR wt (Figure 3E). Thus, the highly stable TAR mutant Del-C4-A16-UCU-22 induced a more potent type I IFN response compared to TAR wt, although not reflected by all measured ISGs and *IRFs*. Strikingly, the TAR mutant Ins-G4-U16-AGA-22, which is even considered more thermodynamically stable, did not enhance type I IFN. These data indicate that deletions that enhance thermodynamic stability, but not insertions, increase antiviral immune responses, suggesting that both stability and the TAR RNA sequence affect viral sensing.

### 2.3. Destabilization of the Lower TAR Stem Enhances Antiviral Immunity

Next, we tested the effect of TAR mutant Substitution AUCA-9 and mutant Substitution UGGU-47, with disrupted base pairing in the lower TAR stem on the induction of antiviral responses (Figure 1C and Figure 2B). Interestingly, the TAR mutant Substitution AUCA-9 significantly enhanced *IFNB* transcription compared to TAR wt-induced responses (Figure 4A,B). Furthermore, a trend towards enhanced *A3G*, *ISG15* and *IRF3* transcription was observed with the Substitution AUCA-9 mutant while leaving *IL27A* and *IRF7* transcription unaffected (Figure 4C,E). The Substitution UGGU-47 mutant enhanced *IFNB*, *A3G*, *IL27A* and *IRF7* transcription, although significance was not achieved due to donor variation (Figure 4B,C,E). *ISG15* and *IRF3* transcription levels were not affected by the Substitution UGGU-47 mutant (Figure 4C,E). The TAR double mutant Substitutions 9 + 47 with restored TAR hairpin integrity (Figure 1C and Figure 2B) induced *IFNB*, ISG and *IRF3* transcription similar to the TAR wt (Figure 4A–C,E), strongly suggesting that the increased IFN response observed for mutant Substitution AUCA-9 and Substitution UGGU-47 is due to loss of structural stability. The single mutants Substitutions AUCA-9 and Substitution UGGU-47 did not affect *IRF7* transcription or resulted in a trend towards increased *IRF7*, respectively (Figure 4D,E). Strikingly, the Substitution 9 + 47 mutant significantly enhanced *IRF7* transcription compared to the TAR wt (Figure 4D,E), suggesting that both TAR stability and the TAR sequence affect antiviral sensing, leading to different immune responses. Our data indicate that lower stability of the TAR structure due to decreased hairpin integrity enhances antiviral immunity, suggesting that the lower TAR stem might be involved in the detection by viral-sensing mechanisms.

### 2.4. PKR Senses TAR Independent of Variation in Stability and Hairpin Integrity

PKR-dependent sensing of abortive HIV-1 RNA (TAR wt) induces NLRP3 inflammasome activation and the secretion of IL-1β in the supernatant of LPS-primed DCs [23]. For the binding of TAR to PKR, the integrity of the TAR hairpin is required [22]. To examine whether hairpin stability and sequence variation in TAR influences PKR-mediated sensing, DCs were treated with LPS or LPS in combination with the different TAR mutants, after which IL-1β secretion in the supernatant was assessed. Treatment of DCs with LPS induced IL-1β background levels, while costimulation of DCs with LPS and TAR wt induced the processing of pro-IL-1β into bioactive IL-1β secreted by the cell (Figure 5). LPS in combination with ATP was used as a positive control for the processing of pro-IL-1β. Strikingly, all different TAR mutants, including the mutants with disrupted hairpin integrity, induced strong IL-1β responses in LPS-treated cells similar to TAR wt-induced responses (Figure 5). Similar to TAR wt, the IL-1β responses induced by the different structures were abrogated in the presence of a PKR inhibitor and unaffected by the DMSO control (Figure 5). Our findings indicate that stabilization of TAR or destabilization of the lower TAR stem does not impede viral sensing by PKR, suggesting that the complete intact TAR stem might not be needed for PKR activity resulting in IL-1β secretion. Furthermore, our data imply that the introduced TAR RNA sequence variation does not affect PKR-dependent sensing.

## 3. Discussion

TAR RNA consists of a 58 nt-long, stably formed hairpin structure present at the 5′ extremity of all HIV-1 transcripts and plays a crucial role in HIV-1 transcription and viral gene expression [1,2,3,10]. TAR present in aborted HIV-1 transcripts results in the induction of antiviral responses via viral sensing mechanisms [2,12,16]. Here, we examined the effect of variation in the TAR hairpin structure and thermodynamic stability on the induction of antiviral immunity and, as such, antiviral sensing. Both stability and TAR RNA sequence affect the induction of antiviral immune responses. Deletions that enhance TAR RNA stability increased antiviral responses, while insertions that further enhanced stability lowered immune responses. Moreover, mutants with a destabilization of the lower TAR stem still resulted in enhanced antiviral responses. Notably, these destabilizing mutants, similar to the stabilizing mutants, elicited pro-inflammatory responses via PKR.

Limited or no variation occurs in the highly conserved three-nt-long bulge element on the 5′ side or the six-nt-long apical loop structure, which is likely due to the crucial role of these regions in Tat-mediated transcription and thus viral replication [6,7,8,9]. It has been shown that changing the structure of the three-nt bulge element by introducing a bulge-complementary sequence enhanced its thermodynamic stability but blocked Tat-mediated gene expression [25]. Interestingly, deletion (Del-UCU-22, delta G −36.2) or insertion of a bulge-complementary sequence (Ins-AGA-22, delta G −41.8) did not abrogate type I IFN responses. These data indicate that, while mutations changing the structure of the three-nt bulge interfere with viral replication, these mutations do not automatically impede viral sensing. These two different processes allow replication-incompetent TAR-containing viruses to still be recognized by viral sensing mechanisms and evoke immune responses. During viral latency, the majority of the proviruses are defective [26,27]. While the activation of viral sensors leads to antiviral immunity and thus interference with HIV-1 replication, triggering of PRRs in these latently infected cells could potentially result in ongoing immune activation and tissue damage. Understanding the role of TAR sequence variation and its effect on viral sensing, therefore, gives insight into both antiviral as well as potential harmful events triggered by these viral sensing mechanisms during non-productive infection and latency.

Our data suggest that the sequence of the TAR RNA mutant Del-C4-A16-UCU-22, possibly in combination with high thermodynamic stability, is favored by PRRs to induce strong antiviral responses compared to the sequence of TAR mutant Ins-G4-U16-AGA-22, while TAR mutant Del-UCU-22 only showed a trend towards enhanced antiviral responses. These data indicate that removal of the UCU sequence at nt position 22 is partially, but not solely, responsible for enhanced type I IFN. Moreover, the distance between TAR and 5′cap might also be a factor leading to enhanced sensing. DDX3 recognizes TAR RNA by the presence of the 5′cap in close proximity to the complex secondary structure, after which type I IFN responses are induced [12,17]. Soto-Rifo et al. have shown that the insertion of an unstructured spacer region elongated the distance from the cap to the 5′UTR, leading to an additional 15 nts, abrogated DDX3-dependent translation of viral RNA [17]. In the TAR mutant Del-C4-A16-UCU-22 the six-nt-long apical loop structure is located closer to the 5′cap compared to the other mutants with a stable hairpin structure, possibly allowing for better recognition by DDX3 and therefore enhancing type I IFN responses. Interestingly, it has been shown that the number of guanosines at the 5′ side of TAR plays an important role in regulating HIV-1 RNA fate in infected cells [28]. The presence of multiple guanosines prolongs the presence of HIV-1 mRNA in these cells [28]. The TAR variants we have developed contain two guanosines at the 5′side of TAR, which might positively affect viral sensing due to prolonged presence in the cytoplasm.

Strikingly, destabilization of the TAR hairpin enhanced antiviral immune responses, suggesting that this is important for viral sensing. It is possible that upon destabilization, more DDX3 molecules can bind to the lower TAR stem or that other parts of the TAR stem become available for other viral sensors, which may enhance antiviral responses. Interestingly, we observed that the Substitutions 9 + 47 mutant increased transcription of *IRF7*, an ISG that is dependent on IFN-dependent JAK/STAT signaling and subsequent activation of ISGF3, and is responsible for the induction of different IFNα species [29]. Therefore, our data suggest that the altered sequence composition of TAR mutant Substitutions 9 + 47 compared to the TAR wt triggers another antiviral pathway.

Recently, we have shown that sensing of abortive HIV-1 RNA by PKR triggers NLRP3 inflammasome activation and subsequent processing of pro-IL-1β [23]. Notably, we observed that both stabilized as well as destabilized TAR mutants induced PKR-dependent IL-1β secretion by DCs. Thus, our data suggest that stabilization of the TAR hairpin is not essential to PKR-dependent sensing. In contrast, it has been described that destabilization of the TAR hairpin abrogates its binding to PKR [22] and that integrity of the TAR stem is crucial for optimal PKR kinase activity [14,30]. Destabilization of the TAR hairpin prevents binding to PKR, however, it does not completely abrogate PKR kinase activity since its residual activity or translational effects could still be detected [14,22]. It is possible that PKR kinase activity and its involvement in inflammasome activation is not dependent on the strong binding of TAR to PKR. Our data suggest that the PKR-dependent sensing mechanism resulting in pro-inflammatory responses is robust as interfering with TAR hairpin integrity did not abrogate IL-1β responses. Notably, mutations in TAR responsible for enhanced type I IFN responses did not automatically result in enhanced IL-1β responses, indicating that those structures could be of interest to specifically boost sensors involved in antiviral immunity. Our data provide a rationale for exploring abortive HIV-1 RNA mutants as adjuvants to specifically enhance antiviral responses while leaving pro-inflammatory responses unaffected.

To conclude, while the mutations we introduced here in TAR RNA likely interfere with the Tat-TAR interactions and negatively affect or even completely abort HIV-1 replication [25,31], this TAR variation can be used to boost viral sensing and, as such, antiviral immunity. Our study shows the robustness of viral sensing mechanisms involved in TAR-dependent induction of antiviral and inflammatory responses.

## 4. Materials and Methods

### 4.1. TAR RNA Sequences of People Living with HIV-1

Sequences of the TAR region of natural HIV-1 subtype B virus isolates were downloaded from the Los Alamos HIV sequence database (last modified November 2020). The 1662 retrieved sequences were aligned and visualized using Weblogo (https://weblogo.berkeley.edu, accessed on 9 July 2021) to assess sequence conservation in silico. Sequences were used to obtain a consensus sequence based on sequence prevalence (Figure 1A).

### 4.2. TAR RNA Mutants

Seven TAR RNA mutants were designed containing different mutations in the hairpin structure. TAR wild-type (wt) was derived from the consensus sequence (Figure 1A) and was used as a reference sequence consisting of the first 58 nucleotides present in the HIV-1 genome (Figure 1B,C) [12,13]. TAR RNA sequences containing the following mutations were designed: Del-UCU-22, Ins-AGA-22, Del-C4-A16-UCU-22, Ins-G4-U16-AGA-22, Substitution AUCA-9, Substitution UGGU-47 and Substitutions 9 + 47 (Figure 1B,C and Figure 2A,B). TAR sequences were analyzed using the Mfold RNA analysis software [32] to predict the RNA structure and Gibbs free energy (∆G) as a predictor of structure stability (Figure 2A,B). All abortive HIV-1 RNA sequence designs were synthesized using in vitro transcription (IVT) and co-capping techniques resulting in RNA structures containing a 5′ m^7^GTP cap and lacking a poly-A tail, as previously described (Bio-Synthesis, Lewisville, TX, USA) [12,13]. Quality control of the TAR transcripts (size and sequence) was performed by Bio-Synthesis.

### 4.3. Cells and Reagents

This study was performed using human buffy coats (Sanquin, Amsterdam, the Netherlands) according to the Amsterdam University Medical Centers, location AMC, Medical Ethics Committee guidelines, and the Declaration of Helsinki. Buffy coats were processed as previously described [33]. In brief, PBMCs were isolated using lymphoprep density separation, followed by a percoll gradient separation to obtain monocytes. To obtain monocyte-derived DCs (DCs), monocytes were cultured in the RPMI supplemented with 10% fetal calf serum (FCS, Biological Industries, Beit-Haemek, Israel), L-glutamine (2 mM, Lonza, Basel, Switzerland), penicillin and streptomycin (100 U/mL and 100 µg/mL, respectively, Thermo Fisher, Waltham, MA, USA) in the presence of GM-CSF (800 U/mL, Invitrogen, Waltham, MA, USA) and IL-4 (500 U/mL, Invitrogen, Waltham, MA, USA) at 37 °C 5% CO_2_ for 6 days. DCs were seeded (1 × 10^5^ cells/well) in a 96-well round bottom plate (Greiner bio-one, Alphen aan de Rijn, Netherlands) and stimulated with 1 nM of the different TAR mutants complexed in transfection reagent lyovec (Invivogen, San Diego, CA, USA). Stimulation of DCs with lipopolysaccharide (LPS) *Salmonella enterica* serotype typhimurium (10 ng/mL, Sigma Aldrich, St. Louis, MO, USA) were performed in the presence or absence of PKR inhibitor C16 (0.5 µM, Merck Millipore, Burlington, MA, USA) or the dimethylsulfoxide (DMSO) control for 2 h (h), or in the presence of recombinant adenosine triphosphate (rATP, 5 mM, Promega, Madison, WI, USA) for the final 4 h of the stimulation.

### 4.4. Quantitative Real-Time PCR

mRNA was isolated using lysis buffer from an mRNA capture kit (Roche, Basel, Switzerland), reversely transcribed to cDNA using a reverse transcriptase kit (Promega, Madison, WI, USA) and PCR amplification was performed with primer sets (Sigma Aldrich, St. Louis, MO, USA) that were developed with Primer Express 2.0 (Applied Biosystems, Waltham, MA, USA, Appendix A) and SYBR green (Thermo Fisher, Waltham, MA, USA) using an ABI 7500 Fast Real-Time PCR detection system (Applied Biosystems, Waltham, MA, USA). The expression level of household gene *GAPDH* was used to normalize the expression levels of target genes: N_t_ = 2^Ct(GAPDH)^^−^^Ct(target)^. Relative expression levels were calculated when N_t_ in TAR wt-treated cells was set at 1.

### 4.5. ELISA

Interleukin 1 beta (IL-1β) secretion levels in the DC supernatant were quantified 24 h after stimulation using Enzyme-linked Immunosorbent assay (ELISA) according to manufacturer’s instructions (Thermo Fisher, Waltham, MA, USA). OD450 nm values were measured using BioTek Synergy HT.

### 4.6. Data Analysis

Statistical analysis was performed using Student’s *t*-test for paired observations using GraphPad version 8. Statistical significance was set at *p* < 0.05.

## Figures and Tables

**Figure 1 pathogens-10-00897-f001:**
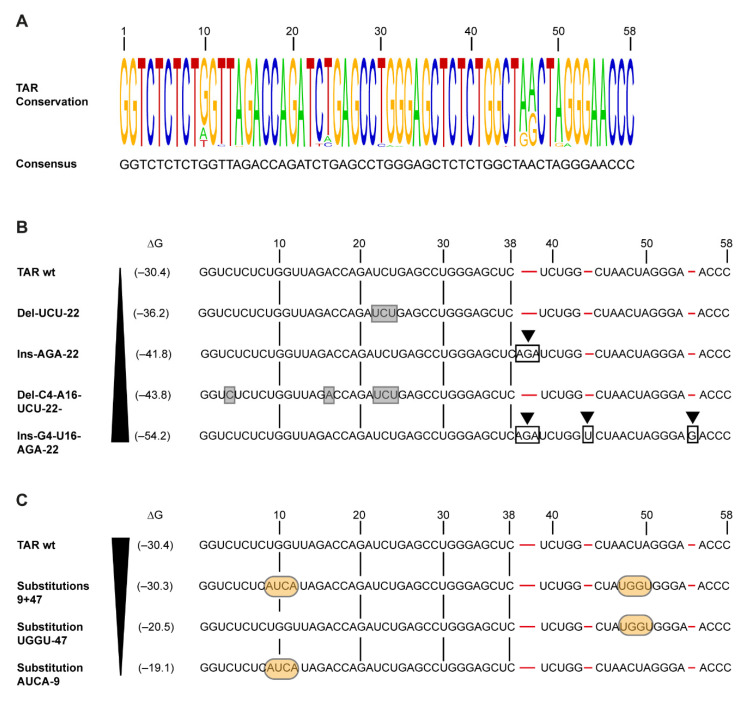
Variation in TAR sequences affects RNA structure and thermodynamic stability. (**A**) Schematic overview of the sequence variation in the TAR region in >1600 natural HIV-1 subtype B virus isolates. Sequences were obtained from the Los Alamos HIV sequence database and aligned using Weblogo, to determine sequence conservation and to obtain a TAR consensus sequence. The size of the depicted nucleotides corresponds to prevalence. Guanine (G, yellow), cytosine (C, blue), adenine (A, green) and thymine (T, red) are shown. (**B**,**C**) Schematic overview of the different mutations in TAR RNA affecting thermodynamic stability (Gibbs free energy (∆G) in kcal/mol) and sequence variation. Deleted nucleotides are boxed in grey, and nucleotide insertions are boxed in black containing an arrow (**B**). Nucleotide substitutions are shown in yellow (**C**). The absence of insertions in other sequences is indicated by a red line (**B**,**C**). Numbers above the TAR wt sequence indicate the nucleotide number of the sequence.

**Figure 2 pathogens-10-00897-f002:**
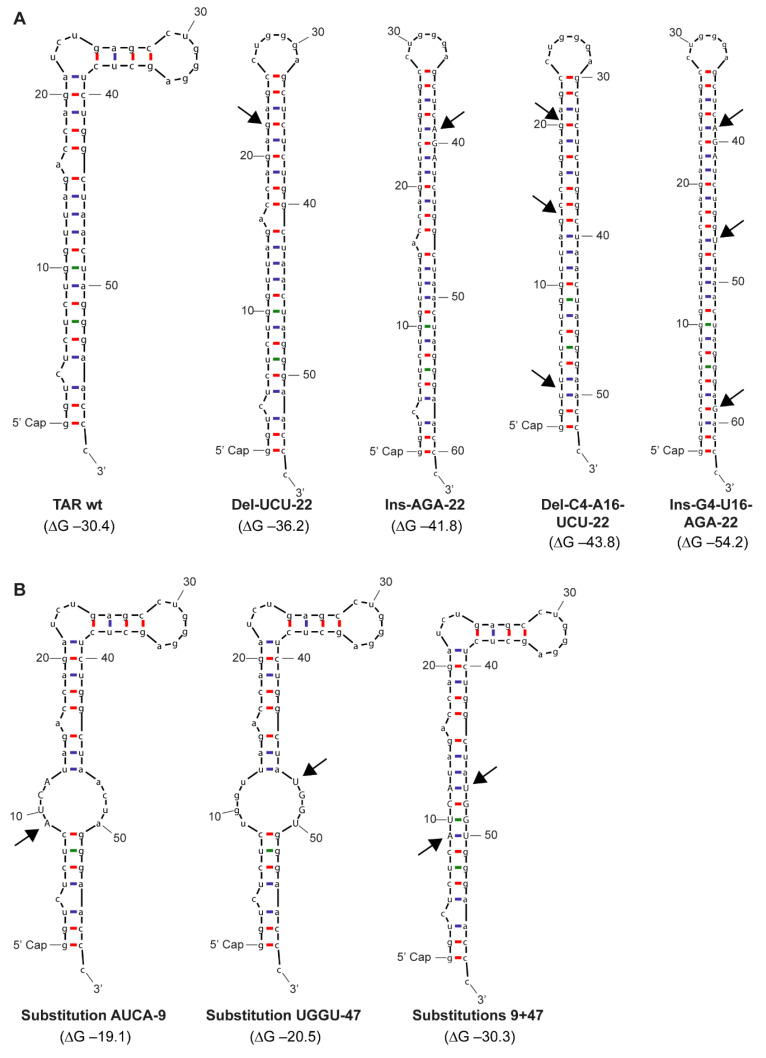
Predicted RNA structure and thermodynamic stability of TAR mutants. (**A**,**B**) Schematic overview of the RNA folding and thermodynamic stability of TAR mutants predicted with Mfold. The thermodynamic stability of the TAR mutants was predicted with Gibbs free energy (∆G) in kcal/mol. A-U bonds (blue), G-C bonds (red) and G-U bonds (green) are indicated. Structures with increasing stability (**A**) or decreasing stability (**B**) are indicated by black arrow. Inserted nucleotides are capitalized.

**Figure 3 pathogens-10-00897-f003:**
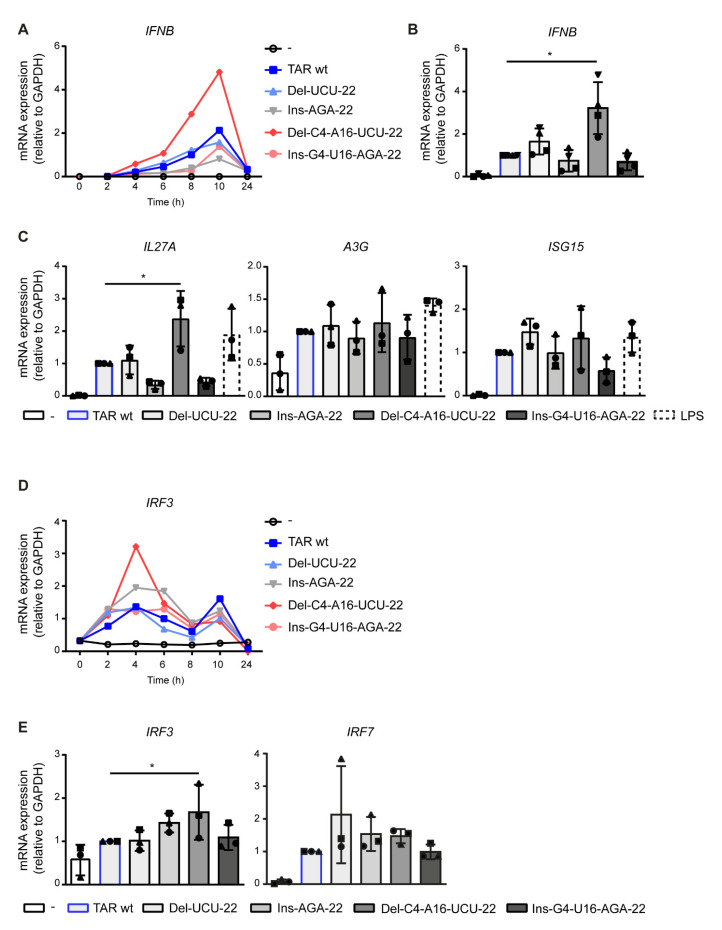
Deletions in the TAR hairpin that enhance its thermodynamic stability increase antiviral immunity. (**A**–**E**) DCs were treated with the different TAR mutants, and *IFNB* and ISGs were measured by quantitative real-time PCR over time (*IFNB* (**A**) and *IRF3* (**D**)) or at designated time points of 8 h (*IFNB* and ISGs (**B**,**C**) and *IRF7* (**E**)) or 4 h after stimulation (*IRF3* (**E**)). mRNA expression was relative to *GAPDH*. Responses induced by TAR wt were set at 1. Data are representative of three donors of different experiments (time courses **A**,**D**) or are collated data of three (**C**,**E**) or four (**B**) donors. Each symbol indicates a different donor (**B**,**C**,**E**). * *p* < 0.05, Student’s *t*-test.

**Figure 4 pathogens-10-00897-f004:**
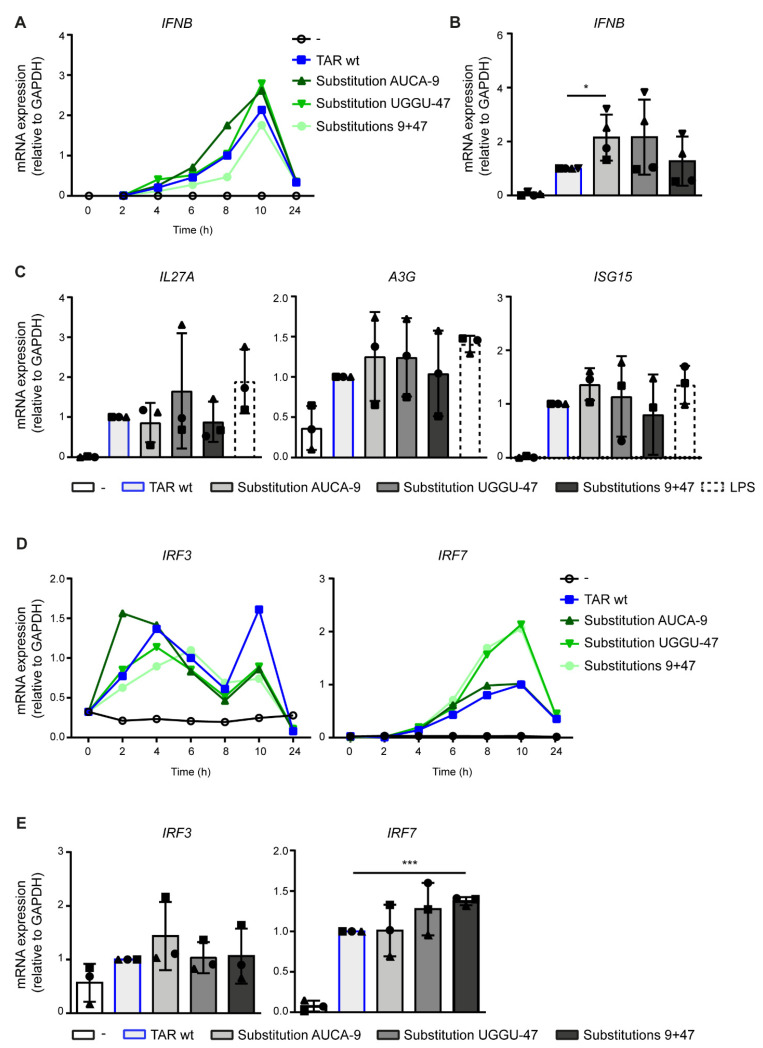
Destabilization of the lower TAR stem enhances antiviral immunity. (**A**–**E**) DCs were treated with the different TAR mutants, and *IFNB* and ISGs were measured by quantitative real-time PCR over time (*IFNB* (**A**) and *IRF3* and *IRF7* (**D**)) or at designated time points of 8 h (*IFNB* and ISGs (**B**,**C**) and *IRF7* (**E**)) or 4 h after stimulation (*IRF3* (**E**)). mRNA expression was relative to *GAPDH*. Responses induced by TAR wt were set at 1. Data are representative of three donors of different experiments (time courses **A**,**D**) or are collated data of three (**C**,**E**) or four (**B**) donors. Each symbol indicates a different donor (**B**,**C**,**E**). * *p* < 0.05, *** *p* < 0.001, Student’s *t*-test.

**Figure 5 pathogens-10-00897-f005:**
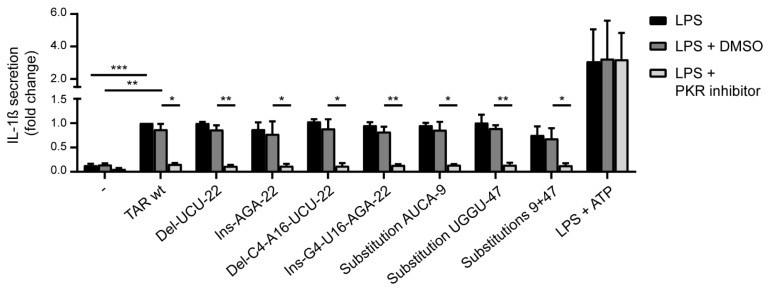
PKR senses TAR independent of variation in stability and hairpin integrity. DCs were treated with LPS, LPS and ATP, or with a combination of LPS and the different TAR mutants, in the presence and absence of PKR inhibitor C16. After 24 h, IL-1β secretion in the DC supernatant was measured using ELISA. Responses induced by the TAR wt were set at 1. Data are collated data of three donors. * *p* < 0.05, ** *p* < 0.01, *** *p* < 0.001, Student’s *t*-test.

## Data Availability

The data presented in this study are available in this article and Appendix A.

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
