# Peer review of "Variations in the Abortive HIV-1 RNA Hairpin Do Not Impede Viral Sensing and Innate Immune Responses"

_pathogens, 2021, doi:10.3390/pathogens10070897_

Round 1

Reviewer 1 Report

In this study, both stabilized and destabilized HIV-1 TAR hairpin mutants were found not to impede antiviral immune responses by monitoring markers such as IFNB, IRF3 and IRF7 etc. The experiments were well designed and the paper was written logically. However, I have following suggestions:

In page 11, 4.2 TAR RNA mutants, authors mentioned that ‘Quality control of structure size was performed using mass spectrometry analysis (Biosynthesis)’. If possible, it is better to include MS data in the supplemental document.

The stability of TAR mutants was only predicted using MFOLD program. Is it possible to validate the stability of variants such as Del-C4-A16-UCU-22 and AUCA-9 compared to TAR wt by performing MS quantification?

In Figure 4E, substitutions 9+47 significantly increased the IRF7 expression instead of IRF3, my suggestion is to add time-dependent curves for IRF7 in Figure 4D.

HIV-1 Tat protein is a known transcription activator binding to HIV-1 TAR RNA to induce HIV-1 transcription. Has the effect of TAR mutants on Tat-TAR interactions been considered or tested already?

Author Response

In page 11, 4.2 TAR RNA mutants, authors mentioned that ‘Quality control of structure size was performed using mass spectrometry analysis (Biosynthesis)’. If possible, it is better to include MS data in the supplemental document.

We apologize for this unclear description. Bio-Synthesis Inc. used MS in the quality control of the TAR RNAs (size and sequence) but not for structure determination. We have clarified the QC by Biosynthesis in the manuscript (line 351-353).

The stability of TAR mutants was only predicted using MFOLD program. Is it possible to validate the stability of variants such as Del-C4-A16-UCU-22 and AUCA-9 compared to TAR wt by performing MS quantification?

Unfortunately, it is not possible to determine the thermodynamic stability by MS.

In Figure 4E, substitutions 9+47 significantly increased the IRF7 expression instead of IRF3, my suggestion is to add time-dependent curves for IRF7 in Figure 4D.

As requested, we have added time-dependent curves for IRF7 in Figure 4D and clarified this in the manuscript and figure legend (lines 192-193, 200).

HIV-1 Tat protein is a known transcription activator binding to HIV-1 TAR RNA to induce HIV-1 transcription. Has the effect of TAR mutants on Tat-TAR interactions been considered or tested already?

We thank the reviewer for raising this point. Indeed, mutations in TAR and the effect on Tat-TAR interaction has previously been investigated (e.g. PMID: 1956776, PMID: 8013464). Although we did not test the effect of the specific mutations described in his manuscript on the Tat-TAR interaction, these mutations will likely negatively affect the Tat-TAR interaction and other TAR functions during viral replication, as the stability and sequence of TAR (in particular the TAR bulge sequence that binds the Tat protein) have been shown to be important for these processes. We have now included a discussion concerning the mutations and effect on replication and included the references (lines 324-326).

Reviewer 2 Report

In this paper Stunnenberg et al studied the effect of TAR mutations on the induction the induction of antiviral and pro-inflammatory responses. Results showed that, whatever the mutations, TAR induces these responses. They can be interesting for people in the HIV field.

The manuscript is well written and relatively short although the discussion is essentially a repetition of the results and can be largely shortened accordingly.

Experiments seem to be well controlled and I just have a few general comments.

Fig.1A is poorly informative because it does not present alternative nucleotides that can be present in up to 30% of TAR sequences (this is the case for A47). The figure would be both more informative and less "old fashioned" if prepared using Weblogo (https://weblogo.berkeley.edu) instead of Jalview.

How strong are the responses elicited by TAR in these different assays? Indeed, it would be helpful to have a positive control such as poly-IC for both the induction of ISGs and IL-1beta.

The authors introduced artificial mutations based on TAR thermal stability but they did not study the effect of the number of guanosines (1 -3) at the 5'-terminus of TAR that was found to control the fate (i.e. mRNA or encapsidation) of HIV RNA in the infected cell (Science 2020; 368 : 413-417). This would have been more biologically relevant. This point should be discussed.

Author Response

The manuscript is well written and relatively short although the discussion is essentially a repetition of the results and can be largely shortened accordingly.

As requested, we have shortened the discussion (lines 242-244, 262-266, 270-271, 287-291, 293-298).

Experiments seem to be well controlled and I just have a few general comments.

Fig.1A is poorly informative because it does not present alternative nucleotides that can be present in up to 30% of TAR sequences (this is the case for A47). The figure would be both more informative and less "old fashioned" if prepared using Weblogo (https://weblogo.berkeley.edu) instead of Jalview.

We thank the reviewer for this suggestion. As requested, we have changed Figure 1A by using Weblogo and have changed the text accordingly (lines 115-117, 334-335).

How strong are the responses elicited by TAR in these different assays? Indeed, it would be helpful to have a positive control such as poly-IC for both the induction of ISGs and IL-1beta.

As requested, we have included LPS as a positive control for ISG induction (Figures 3C and 4C), and used LPS + ATP as a positive control for IL-1beta secretion (Figure 5). We have clarified this in the manuscript (lines 147-148, 215-216, 368-370).

The authors introduced artificial mutations based on TAR thermal stability but they did not study the effect of the number of guanosines (1 -3) at the 5'-terminus of TAR that was found to control the fate (i.e. mRNA or encapsidation) of HIV RNA in the infected cell (Science 2020; 368 : 413-417). This would have been more biologically relevant. This point should be discussed.

We thank the reviewer for raising this point and have included and discussed the effect of the number of guanosines in the discussion (lines 282-286).